# A model for time interval learning in the Purkinje cell

**Daniel Majoral**[1,2]*, **Ajmal Zemmar**[1,3], **Raul Vicente**[1,2]*

**1** Department of Neurosurgery, Henan Provincial People's Hospital of Zengzhou University, School of Clinical Medicine, Henan University, Zengzhou, Henan, China, **2** Computational Neuroscience Lab, Institute of Computer Science, University of Tartu, Tartu, Estonia, **3** Department of Biology and Department of Health Sciences and Technology, ETH Zurich, Zurich, Switzerland

\* danielmajoral@gmail.com (DM); raulvicente@gmail.com (RV)

**Data Availability Statement:** All relevant data are within the manuscript and its Supporting Information files.

**Funding:** RV and DM thank the financial support from the Estonian Research Council through the personal research grant PUT1476

## Abstract

Recent experimental findings indicate that Purkinje cells in the cerebellum represent time intervals by mechanisms other than conventional synaptic weights. These findings add to the theoretical and experimental observations suggesting the presence of intra-cellular mechanisms for adaptation and processing. To account for these experimental results we propose a new biophysical model for time interval learning in a Purkinje cell. The numerical model focuses on a classical delay conditioning task (e.g. eyeblink conditioning) and relies on a few computational steps. In particular, the model posits the activation by the parallel fiber input of a local intra-cellular calcium store which can be modulated by intra-cellular pathways. The reciprocal interaction of the calcium signal with several proteins forming negative and positive feedback loops ensures that the timing of inhibition in the Purkinje cell anticipates the interval between parallel and climbing fiber inputs during training. We systematically test the model ability to learn time intervals at the 150-1000 ms time scale, while observing that learning can also extend to the multiple seconds scale. In agreement with experimental observations we also show that the number of pairings required to learn increases with inter-stimulus interval. Finally, we discuss how this model would allow the cerebellum to detect and generate specific spatio-temporal patterns, a classical theory for cerebellar function.

## Author summary

The prevailing view in neurosciences considers synaptic weights between neurons the determinant factor for learning and processing information in the nervous system. Recent experimental findings indicate that Purkinje cells in the cerebellum represent time intervals by mechanisms other than conventional synaptic strength. We propose a new biophysical model, which complements the modification of synaptic weights, to account for the learning of time intervals in one dendrite of the Purkinje cell. The model is based on calcium oscillations in microdomains and learns inter-stimulus intervals in the sub-second and second range. Our findings provide insights regarding the mechanisms for delays in cerebellum circuitry, which allow to detect and generate specific patterns in space and time, a classical theory of cerebellar function.

(https://www.etis.ee/Portal/Projects/Display/52ed4301-f2ef-4364-9770-397e31936f93?lang=ENG) and the Estonian Centre of Excellence in IT (EXCITE) funded by the European Regional Development Fund, through the research grant TK148 (https://www.etis.ee/Portal/Projects/Display/fd0aeffa-a7d3-4191-b468-0f44aa2847af?lang=ENG). This work was supported by grants of the Henan Provincial People's Hospital Outstanding Talents Founding Grant Project, the Heidi Demetriades Foundation, UZH Forschungskredit (FK 15-5), EMDO Foundation (872), the ETH Foundation and the Swiss National Science Foundation (grant IZK0Z3-150809) to A.Z. The funders had no role in study design, data collection and analysis, decision to publish, or preparation of the manuscript.

## Introduction

The brain's ability to measure time, discriminate temporal patterns, and produce appropriately timed responses is critical to many forms of learning and behavior [1]. This is evident in the cerebellar system's involvement in the learning and expression of the timing of associations by classical conditioning, such as delay eyeblink conditioning [2]. However the exact mechanism through which the cerebellum and related structures exert this function has been a matter of debate [3–5]. Many studies have tried to locate the constituent of the cerebellar system that learns and stores the time intervals during delay eyeblink conditioning [6].

In delay eyeblink conditioning a previously neutral stimulus, the conditioned stimulus or CS (e.g. auditory cue) is paired with another stimulus, the unconditioned stimulus or US (e.g. air puff to the eye) that by itself produces a conditioned response or CR (blink). The US is presented with a certain time delay from the CS. After repeated exposure this time delay to the onset of US is learned and when the CS is presented alone, the CR appears slightly before the expected US. The main circuitry for delay eyeblink conditioning seems to be comprised of (see Fig 1): parallel fibers that convey information about the CS (e.g. auditory cue); climbing fibers that are activated by the US (e.g. air puff to the eye); and the inhibitory Purkinje cells, the output elements of the cerebellar cortex, which seem to be responsible for gating the CR (blink). Thus the timing mechanism might be located in the cerebellar cortex [7].

Following on this line of research, Johansson and colleagues [8] were recently led to the conclusion that the timing information for the conditioned responses is stored in the Purkinje cells of the cerebellar cortex, and that the intervals were not stored as usual synaptic weights, but as some other yet to be understood mechanism. To isolate the element storing the timing information, Johansson and colleagues [8] used direct electrical stimulation of the Purkinje cell afferents (climbing and parallel fibers), bypassing the other elements of the cerebellar system. Mimicking the delay eyeblink conditioning protocols, the electrical stimulation of the climbing fibers operated as the US; the stimulation through the parallel fibers acted as the CS; and the Purkinje cell pause in firing was the response (CR). The success of this approach is

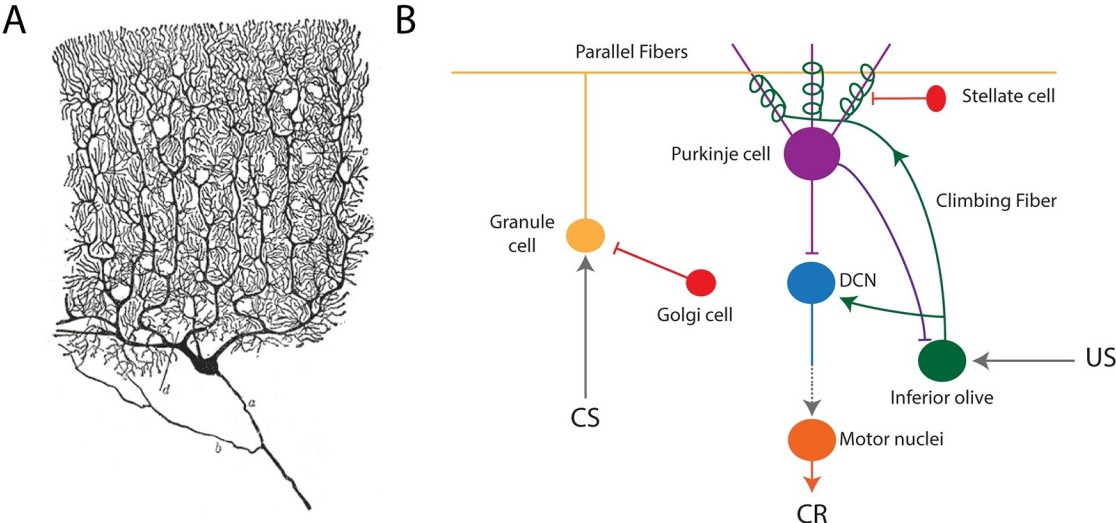

**Fig 1. Basic cerebellar circuitry.** (A) Drawing of a Purkinje cell in the cat's cerebellar cortex, by Santiago Ramón y Cajal. (B) Basic circuit in eyeblink conditioning. The Purkinje cell receives the conditioned stimulus (CS) through parallel fibers from granule cells. The unconditioned stimulus (US) arrives through the climbing fiber from the inferior olive. The Purkinje cell also receives inhibitory inputs from stellate cells. Purkinje cell output is inhibitory to the deep cerebellar nuclei (DCN), which by rebound firing activates motor neurons generating the conditioned response (CR).

attested by the fact that after training, the electrical CS is enough to generate the CR (maximal amplitude of Purkinje cell pause in firing) before the expected onset of the US, and return to firing baseline just after the expected US [8].

These and other experiments show that partial stimulation of the parallel fibers (interrupted CS) still generates the CR at the same learned interval [9]. Also a change in stimulation frequency still induces the CR [10] at the expected time interval. If synaptic weights (between parallel fibers and Purkinje cell) would code for the timing, the expressed interval would depend on the drive duration and frequency. Therefore there seems to be a decoupling of the timing mechanism from further drive to the synapse. Interestingly the blockade of the main ionotropic glutamate receptors (NMDA and AMPA) did not prevent the expression of the conditioned responses or their timing [11], while blocking mGluR7 (Metabotropic Glutamate Receptor 7) prevented the timed response. mGluR7 is a G-protein coupled receptor, and hence the binding of glutamate to the receptor does not open a channel immediately, as it would be the case for ionotropic receptors, but initiates a molecular cascade that hypothetically can hyperpolarize the cell. More experiments are necessary to confirm and detail these findings, but a cascade initiated by the activation of mGluR7 is a candidate mechanism for decoupling the learning of timing from further synaptic activity.

In this work we propose a biophysical model to explain these observations on time interval learning in Purkinje cells. In particular, we propose a model for the short time scale of the process before a long term memory is consolidated. Our model is based on known pathways invoked to explain other forms of classical conditioning [12]. For example, elements of our model such as a form of adenylyl cyclase (Rutabaga) has been shown to act as a coincidence detector and activator of the cAMP-PKA pathway in drosophila olfactory classical conditioning [13]. The same pathway also underlays classical conditioning of the gill-withdrawal reflex in Aplysia [14].

From a computational perspective the model relies only on a few steps which can be readily implemented via existent pathways in Purkinje cells. In a specific implementation CS from parallel fibers generates a calcium signal in a small domain that keeps track of the time elicited since the synaptic activation. The arriving US from the climbing fiber liberates a G protein. The coincidence of calcium release and G protein liberation activates adenylyl cyclase, which in turn changes the calcium signal and the activation time of calcium-dependent proteins that lead to the cell hyperpolarization. After several CS-US pairings we show how the proposed cascade makes the timing of the Purkinje cell inhibition converge to the correct time interval.

## Results

### Model

We first proceed to explain the computational properties of the model and then propose a biophysical implementation. Let us first consider a system that receives an input at some point in time. Later, through a different channel, the same system receives a second input and it should be able to store the interval between the onsets of both signals.

To store such timing information and satisfy experimental and biophysical constraints the system only needs a small set of mechanisms. First, a variable in the system should reflect the information that an input arrived and do so independently of the duration and strength of such input. To do so the system must posses a mechanism that elicits the same output even for different input strengths, and we call this element a decoupling mechanism. The output of the decoupling mechanism is a time dependent process that measures time, and we call such a process "clock". The coincidence of an active "clock" with the arrival of a second input is registered by a coincidence detector. A final step is that the coincidence detector feeds back to the

"clock" to regulate its "speed". Positive and negative feedback loops can ensure that elements integrating the "clock" signal and acting as an output will activate at the correct timing. See Fig 2 for the proposed network of interactions able to learn time intervals.

The proposed model contains five main elements: a decoupling mechanism, a "clock", a coincidence detector, and a positive and a negative feedback loop. These elements can be implemented by one or several molecules or alternatively one molecule can implement several of the mechanisms. Next we discuss the biophysical level of the model and identify some possible candidates for these mechanisms:

1. Calcium as a "clock". Calcium is one of the most common messengers in cells with a wide range of roles. Interestingly one of these roles is timing [15]. Multiple molecules activate depending on the spatio-temporal profile of the calcium signal [16, 17]. In our model the integration over time of the calcium signal allows the adjustment to the inter-stimulus interval.

2. Inositol trisphosphate (IP3) and calcium as a decoupling mechanism. The parallel fiber signal generates IP3 which will couple to receptors in calcium stores, releasing calcium. The first milliseconds of electric signal generate enough IP3 to saturate the receptors, thus decoupling the released calcium from prolonged electrical stimulation. Once the calcium is released a CIRC process (Calcium Induced Calcium Release) [18] makes the calcium signal continue independently of the incoming electric signal.

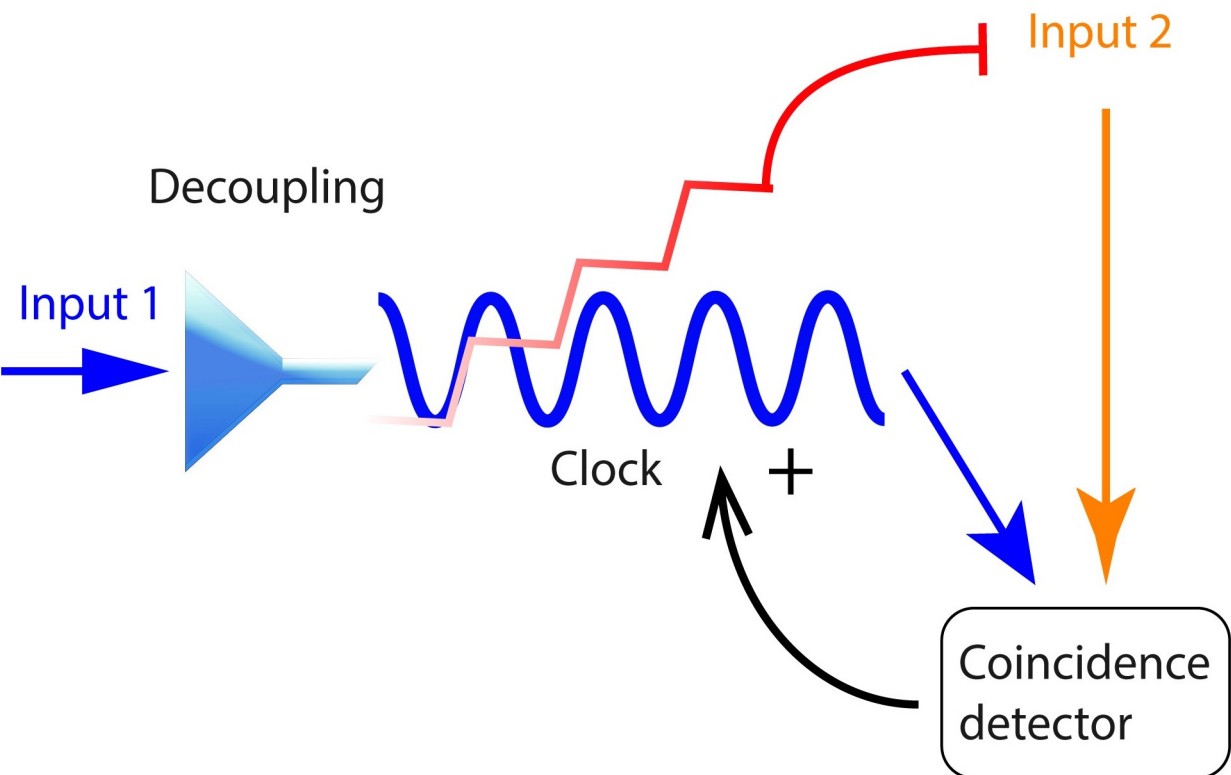

**Fig 2. Model scheme.** Basic elements of a computational system that is able to reproduce the experimental results in delay eyeblink conditioning for the Purkinje cell. A first input goes through a decoupling mechanism and activates a clock. If the clock accumulated activation coincides with another signal a coincidence detector is activated. The coincidence detector exerts a positive feedback that acts on the clock. A negative feedback activated by the clock acts on the second signal, if it gets activated quick enough it deactivates the second input.

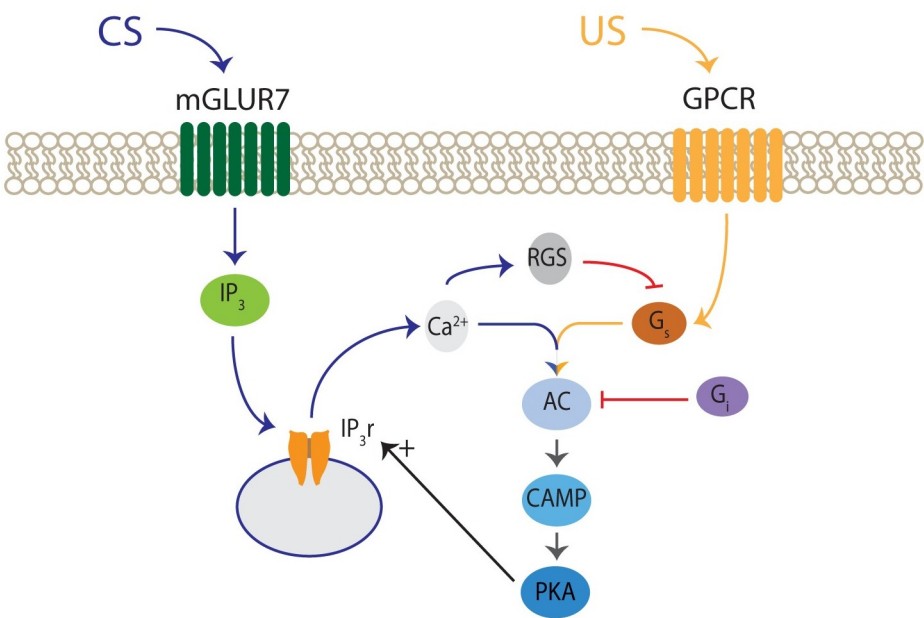

**Fig 3. Scheme of molecular interactions in the model.** The opening of IP3 receptors (IP3r) liberates calcium (Ca2+). Both calcium (Ca2+) and G alpha stimulatory protein (Gs) stimulate adenylyl cyclase in a synergistic manner. AC generates cAMP which activates PKA. PKA creates a positive feedback opening more IP3 receptors. Calcium also activates RGS proteins wich create a negative feedback through inhibition of Gs.

3. Adenylyl cyclase as a coincidence detector. Adenylyl cyclase (AC) is sensitive to calcium/calmodulin and stimulatory G protein [19]. The coincidence of calcium and stimulatory G protein activates AC more than the sum of individual activations. This synergistic activation is also dependent on the order of arrival of the stimulus. When the calcium signal precedes the G protein the activation is more effective than in the reverse order [19]. These attributes of adenylyl cyclase account for some aspects of delay eyeblink conditioning, in which the CS has to precede the US to produce associative learning.

4. Protein kinase A (PKA) as a positive feedback. Adenylyl cyclase activates PKA which in turn phosphorylates the IP3 receptor in calcium stores. This facilitates the release of calcium [20]. The increased release of calcium increases the activation of adenylyl cyclase and PKA.

5. Regulators of G protein signaling (RGS) as a negative feedback. Some forms of RGS are activated by calcium and in turn diminish the activation of adenylyl cyclase. Thus, when RGS is increasingly activated with each pairing, the activation of AC will diminish. After RGS activation compensates the positive feedback loop, further training at the same timing will not alter the level of AC. This negative feedback loop ensures the convergence of AC levels and the timing of calcium-activated proteins.

The biochemical actors proposed in the model interact in dendrites of a Purkinje cell. The diagram shown in Fig 3 summarizes the involved molecular pathways and their interactions. In particular, input from parallel fibers causes the release of glutamate and activation of mGluR7 receptor. The activation of mGluR7 activates inhibitory G protein (Gi). The activation of mGluR7 also leads to the formation of IP3, which bounds to IP3R channels in the endoplasmic reticulum, releasing calcium from internal stores. The calcium signal can continue for a few seconds due to calcium induced calcium release mechanisms. Calcium alone weakly stimulates some forms of adenylyl cyclase and inhibits others. However if a second input

generates stimulatory G protein (Gs), adenylyl cyclase is strongly activated. AC initiates the cAMP-PKA signaling pathway which exerts a positive feedback on the calcium signal. Finally, calcium dependent RGS inactivates G protein-coupled receptor pathways performing a negative feedback loop.

In the Methods section we provide a detailed account of the the numerical simulation of the kinetics of calcium and all involved proteins in Fig 3 during their interactions.

**Model dynamics after a single CS-US pairing.** In the preceding section we have seen all the elements that compose the model. Here we show how they react when a CS is paired with a US. When a signal arrives at the synapse (through parallel fibers) it first activates mGluR7 receptors, mGluR7 activation generates IP3 through a molecular cascade and opens intra-cellular calcium stores near the synapse. Thus the conditioned stimulus (CS) causes a sustained calcium signal as seen in the numerical simulation of the model shown in Fig 4A. The US stimulus from the climbing fiber activates a receptor in the cell liberating Gs alpha protein. The coincident presence of calcium (CS) and Gs alpha protein (US) induces AC activation [19] just after US arrival as shown in Fig 4B. In turn elevated AC levels increase the synthesis of cAMP, and PKA level is observed to rise smoothly until reaching a stationary level as seen in Fig 4C. Finally, PKA phosphorylates the IP3 receptors and increases the release of calcium [20]. Thus, when tested with a second CS-US pairing, the level of calcium released by CS has increased (Fig 4D). In our model comparing the two pairings (Fig 4A and 4D) the second pairing induces around 100% more calcium release.

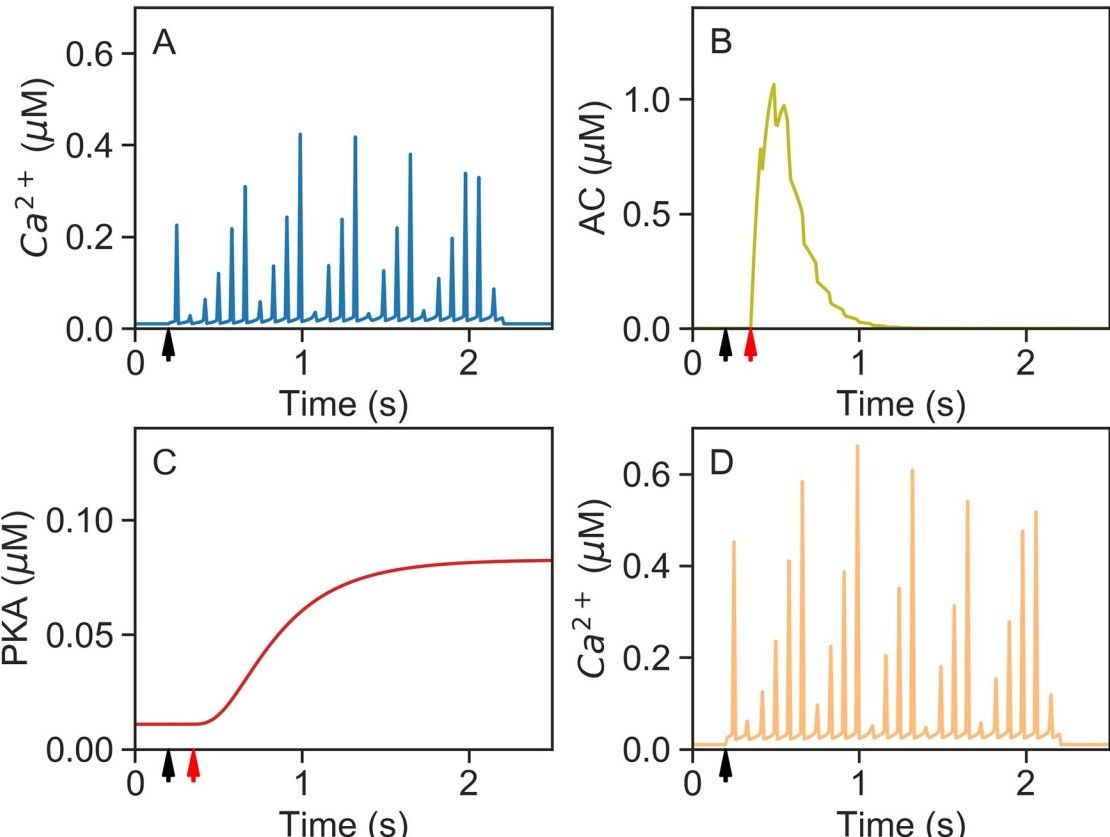

**Fig 4. One pairing of CS-US with 150 ms inter-stimul us interval.** Black arrow: parallel fiber input. Red arrow: climbing fiber input. (A) Calcium released for one CS-US pairing, (B) adenylyl cyclase evolution, and (C) PKA level, during the first pairing. (D) Calcium released during a second pairing.

**Learning one time interval.**    We have seen that a single CS-US pairing changes the activation of AC and PKA level, and increases the calcium elicited upon the arrival of the next CS. In this section we study in more detail how such dynamics allows the synapse to learn one time interval.

In our biophysical model a key to learn the inter-stimulus timing and perform the appropriate response are regulators of protein signaling (RGS). RGS proteins can vastly increase Gi alpha protein activity which inhibits AC [21]. Some forms can also inhibit Gs alpha protein production which stimulates AC, or inhibit AC by allosteric regulation. RGS proteins are inactivated by phosphatidylinositol-3-phosphate (PIP3) but calcium nullifies this inactivation [22].

In the model, before pairing of CS-US occurs, RGS proteins are inactivated by PIP3 and the basal level of calcium is not enough to activate them. When the CS arrives the calcium signal will activate RGS proteins by cancelling PIP3 inactivation. During successive trainings the higher level of PKA increases the calcium signal elicited by a CS. Thus, RGS proteins will be activated earlier with each pairing, until RGS proteins are activated just before US arrival, and as a result inhibiting the Gs activation of AC. In Fig 5 we plot the dynamics of RGS response and its dependence on the number of pairings (each with a CS-US time interval of 150 ms). It can be clearly seen that RGS protein responds abruptly to the input signal for the first pairing, but with a one second delay with respect to the climbing fiber input. Posterior pairings reduce the time delay, and as shown for eleven pairings the RGS protein already anticipates the climbing fiber signal.

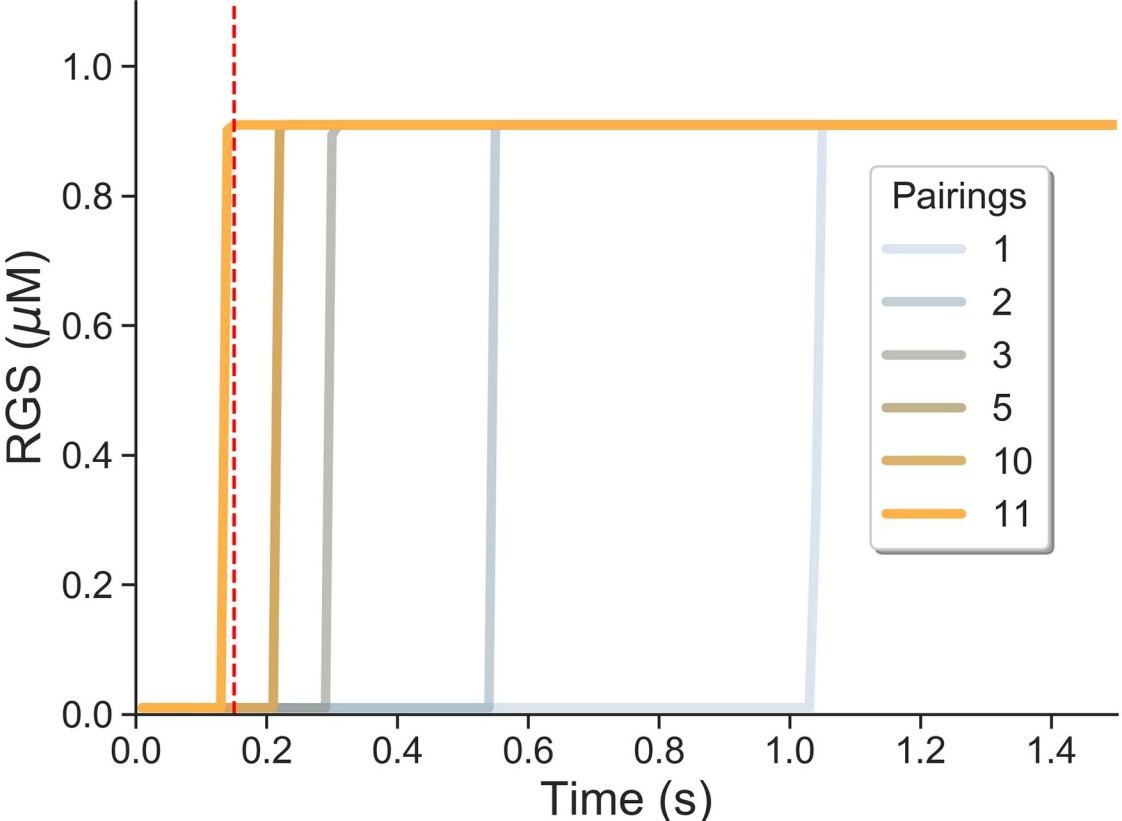

**Fig 5. RGS protein learns to anticipate climbing fiber signal.** Response of the model to delay eyeblink conditioning with an interstimulus interval of 150 ms. Red dashed line indicates the arrival of the climbing fiber input. Time course of RGS protein activation depends on the number of previous pairings. After training with 11 pairings protein activation anticipates climbing fiber input.

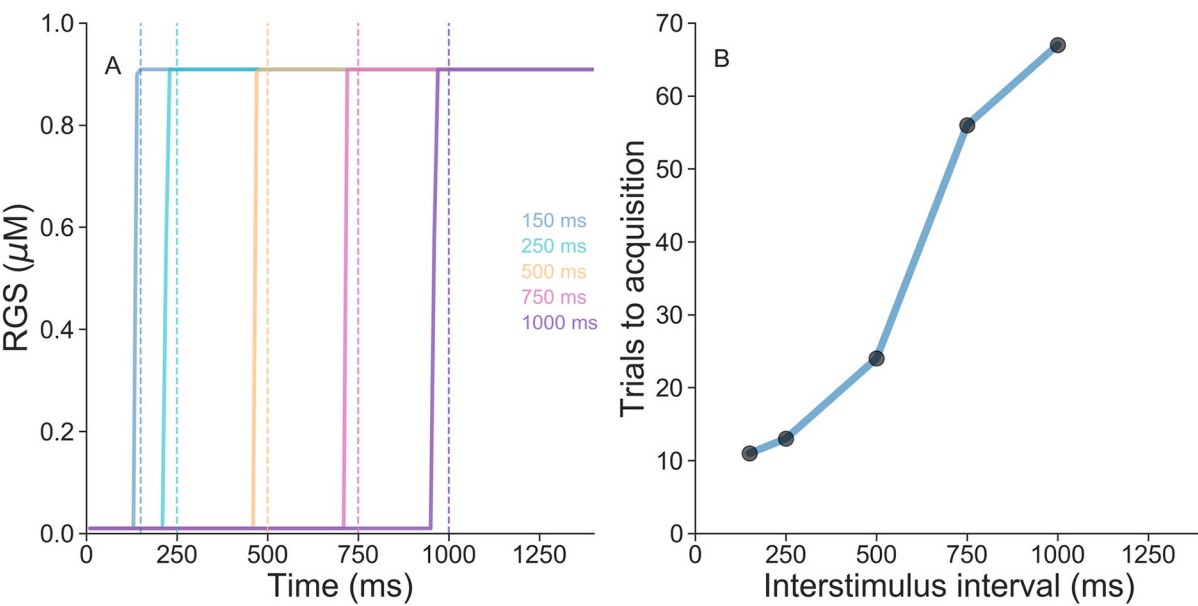

**Fig 6. RGS protein anticipates CF signal for different time intervals.** (A) Dashed line indicates the arrival of the climbing fiber input for the different inter-stimulus intervals. Continuous line shows the activation of RGS protein activity for the different inter-stimulus interval after 70 CS-US pairings. (B) Required number of pairings to learn a timed response for the different inter-stimulus intervals.

**Learning different time intervals.** We have seen how the dynamics of the molecules in the model self-organize to represent a 150 ms time interval by shifting the activation of some proteins toward the inter-stimulus interval. In this section we extend the result to a more broad range of time intervals and point out some differences between short time intervals versus longer ones.

We proceeded by training the model for several inter-stimulus intervals in the range 150-1000 ms. That is, we stimulated the model with paired CS-US inputs with different time intervals. As shown in Fig 6A after 70 pairings the activation of RGS proteins learns to anticipate the climbing fiber signal for all cases. In simulations with longer time intervals RGS activations similarly anticipated the climbing signal indicating than learning can occur beyond the subsecond time scale. We also measured the number of pairings needed for the RGS activation to anticipate the US as a function of the inter-stimulus interval. Fig 6B shows that the necessary number of pairings increases with the inter-stimulus interval in line with experimental observations [23].

## Discussion

Recent experimental results in delay classical conditioning [8, 11, 24] indicate that some form of time interval learning takes place inside the Purkinje cells in the cerebellum. Here we have developed a biophysical model for time learning in dendrites of the Purkinje cell. In the model parallel fibers input (CS) elicits a calcium signal followed by climbing fiber input (US) activating G protein coupled receptors that liberate Gs protein. The coincidence of calcium and Gs protein activates adenylyl cyclase, which via a PKA cascade changes the calcium signal and the activation time of calcium-dependent proteins. After a few CS-US pairings, the activation time of calcium-activated proteins learns to peak at the interstimulus interval and the Purkinje cell can pause at the correct time.

The general idea proposed here is that, regardless of the specific molecules involved, molecular pathways inside dendrites of Purkinje cell possess the necessary machinery to learn time

intervals. Indeed the computational steps leading to time interval learning and explaining recent experimental observations are simple, and only involve basic interactions such as a saturating response, a coincidence detector and a negative feedback loop.

## Relation to other models of time interval learning in Purkinje cells

The model presented here has similarities to the model proposed by Steuber and Willshaw [25], both using a calcium signal acting as a memory trace to link the timing between two events. However their model is very similar to classical synaptic plasticity mechanisms since it proposes that the time learning modifies the availability of mGlu receptors. Therefore their model lacks a mechanism that decouples delay timing from electric strength which is in contradiction with experimental results. Our model considers the number of receptors fixed, thus if their response saturates for small short signals the delay is independent of electric strength and duration.

Including the lack of decoupling mechanism Johansson and colleagues [10] pointed out several more shortcomings of the model proposed by Steuber and Willshaw:

1. The model learns for short interstimulus intervals(<100 ms) in contradiction with experimental data [26].

2. If the interstimulus interval is changed in the model the pause response will move gradually to the new time. Experimentally the old time response disappears and a new response appears without passing through intermediate delays [8, 9].

3. A Purkinje cell trained with two interstimulus intervals generates two responses [8, 9]. The model is only able to generate one response not two responses for two different times.

In our model the first problem is absent due to AC activation profile which prevents the learning of very short time intervals. The activation of AC is highly dependent on the temporal ordering of stimuli: if the Gs protein (elicited by US) is released before calcium (elicited by CS) or simultaneously the response of AC is small. Given that it takes some time for the calcium signal to start, for short interstimulus intervals the learning will not take place.

In line with Jirenhed et al. [27] we would also like to suggest a simple explanation to avoid the second and third shortcomings; given that the Purkinje cell might have around 100000 synapses the learning can take place at many synapses in a stochastic manner. Different synapses can learn different times producing more than one timed response. Also the transition to a new response will not pass through intermediate time intervals because new synapses will learn the new time meanwhile the old time is erased from others.

The capability to learn at multiple dendrites offers additional advantages. For example, in a natural situation the Purkinje cell would have to recognize sparse incoming sequences. The ability of a Purkinje cell to learn timing information at multiple synapses helps to integrate evidence arriving at different times and produce appropriate responses. In other words, even though neurons could be modeled with a single delay mechanism that learns more than one time to match the experimental evidence obtained in Purkinje cells, it is more likely that in natural conditions the synapses would have to be timed independently to produce the behavioral response at the adequate time. Moreover the later is able to explain better other cerebellum functionality beyond delay conditioning as we suggest below.

## Implications for classical theories of cerebellar function

The cerebellar anatomy seems well understood, to a first approximation it consists of a single circuit that repeats itself [28]. However many functions have been assigned to the cerebellum

[29] i.e.: timing of classical conditioning [30]; prediction of stimuli [31]; error correction [32]; forward models in motor control [33] and motion compensation in perception [34]. The lack of a single function for the cerebellum seems to be at odds with its structural uniformity. However, some proposals suggest that its structural uniformity reflects the computations performed by the cerebellum [35], even if the higher level functions that exploit these computations look radically different.

One of the first theories for cerebellum function is the proposal by Braitenberg [36], in which the cerebellum responds specifically to certain time sequence of events in the input and learns to generate other specific sequences of signals in the output. Originally, it was proposed that the key to achieve this computation was the existence of different delays or latencies of the input to Purkinje cell, but given the lack of delay lines in cerebellum circuitry Braitenberg modified his hypothesis [37]. In particular, the lack of experimental evidence of delays in cerebellum circuitry made very difficult to justify how the cerebellum might integrate past information to learn time sequences.

However, it is interesting to consider Braitenberg old hypothesis given experimental findings in delay eyeblink conditioning [8]. It is possible that delay eyeblink conditioning can be reproduced with the Purkinje cell learning time intervals at the soma, but to explain other cerebellum functions the Purkinje cell has to be able to integrate time sequences. If each Purkinje possesses multiple timing mechanism along the dentritic tree, it will allow the cell to respond to complex spatio-temporal inputs with any desired output. Normally there will be a cause-effect relation between inputs and output: the output might be a motor action to avoid or seek the effects of the inputs, or the effects themselves to be processed in another brain region (one possible exception is that the output is the timing between two inputs). The main limitation of this approach is the time between input and output, the further away in time the more difficulty is to model cause-effect given that the possibles causes are all the combinations of all the inputs received. Thus for learning to predict there is a trade-off between how much time into the future the prediction goes and the number of possible causes taken into consideration. Cerebellum seems well posed to favor the later, that is, processing sensorimotor information and making predictions in the millisecond range.

## Model limitations

Our main contribution is the proposal of a computational model able to self-organize to learn appropriate time intervals. However, we also explicit out a possible molecular implementation of the model. We believe there are many alternatives to such a implementation and indeed here we discuss several limitations of our own specific proposal.

- Many parameters of the model have been obtained from in vitro studies. Typically in vitro assays for biological reactions are studied in equilibrium and in controlled conditions. However biological pathways might depend on the dynamics of the signals [38] and the convergence of multiple interactions. In our case AC activation might depend on the timing between CS-US, this dependence can differ greatly from the model, depending on the conditions inside the cell.

- Here we have modeled that the US arriving from climbing fibers activates G protein coupled receptors (GPCRs) without entering in detail. Given that the climbing fiber innervates all the dendritic tree, it might also activate receptors in a more direct manner to elicit the learning mechanism. Another possibility is that the pause in firing activates the receptors. However, optogenetic essays in the Purkinje cell [39] show that a pause alone does not suffice to learning the timing: instead when increased excitation in Purkinje neurons precedes a pause

(without teaching signal from climbing fiber), the timing is learned. It is possible that when a climbing fiber signal arrives, the high frequency burst of somatic spikes activates a retrograde chemical signal that activates GPCRs along the dendritic tree.

- We have modeled interval timing with a calcium signal. Although it is known that calcium signals and oscillators can reflect timing information, is not clear how such information is related to the spatio-temporal profile of the calcium levels. In our model RGS proteins respond mostly to the amplitude of calcium over time, however another possibility is that their activation depends on Ca2+ frequency or a specific number of pulses [40, 41]. This would make the read-out more robust to changes in basal Ca2+ levels. Also to make the model simpler we have modeled activation of RGS proteins almost instantaneous after certain level of calcium, but this might differ greatly in reality.

- We have oversimplified the signal coming from the inferior olive through the climbing fiber and how it affects the Purkinje cell. Also we did not take into account the feedback loop between Purkinje cell and inferior olive [42]. For example, the inferior olive with its own timing mechanisms [43] might compute time differences between Purkinje cell output and US, perhaps changing the teaching signal in accordance. Our model is a simple proof of principle for intracellular models of timing mechanisms and does not exclude neither informs of other mechanisms that probably exist in the cerebellar system.

## Conclusion

We have shown that is possible to learn time intervals at a single dendrite without classical synaptic plasticity. Although we expect the real mechanism to be more complex, the model is biologically plausible and serves to illustrate that a few computational steps can endow Purkinke cells with time interval learning abilities. The model is based on the cAMP-PKA pathway involved in several forms of classical conditioning. In the model a calcium signal keeps track of time, similarly to other mechanisms of biological timing, and the "speed" of the signal is self-adjusted via feedback loops until calcium-dependent proteins activate at the appropriate time interval.

The capacity to learn time intervals at each one of the dendrites in the Purkinje cell might validate Braitenberg theory: the cerebellum responds to certain time sequences of events in the input and learns to generate other specific sequences of signals in the output. Future experimental evidence will be needed to bridge the gap between molecular mechanisms at microdomains and psychological function in the cerebellum.

## Methods

Simulations were performed in the NEURON environment (Version 7.4) [44], with backward Euler integration method and 25 $\mu$s time steps. The simulations where executed at the Comet supercomputer at the San Diego supercomputing center through the Gateway Portal [45].

### Calcium store

The calcium oscillations follow the Somogyi-Stucki Model [46]:

$$\frac{dx}{dt} = k_1 y - k_2 x - \alpha f(y) x,$$

(1)

$$\frac{dy}{dt} = k_2 x - k_1 y + \alpha f(y) x + \gamma - \beta y,$$

(2)

$$f(y) = \frac{y^n}{y^n + h^n}, \tag{3}$$

where x is the calcium concentration inside the pool and y in the cytosol. When there is no parallel fiber stimulation the calcium at the synapse is at certain basal level:

$$[Ca^{2+}] = B_{Ca}. \tag{4}$$

PKA phosphorylation induces at least a fourfold increase in the sensitivity of InsP3R1 to activation [47]. Therefore when a parallel fiber stimulus arrives we initiate the value for the calcium inside the pool at 25 (x = 25), and for two seconds (before returning to basal level) the calcium at the synapse follows the formula:

$$[Ca^{2+}] = (1 + 10[PKA])y, \tag{5}$$

where [PKA] is the level of active PKA and $[Ca^{2+}]$ the calcium released.

## Gi alpha subunit

mGluR7 activation increases Gi during 1.2 seconds:

$$G_i = ae^{bt}, \tag{6}$$

where t is the time since mGluR7 activation. After two seconds the Gi alpha subunit is deactivated.

## PKA pathway

Active adenylyl cyclase is given by the following equation:

$$\frac{d[AC]}{dt} = r_1[AC_i] - r_2[AC]. \tag{7}$$

To take into account the synergy between calcium and G protein to activate adenyl cyclase we have taken the following rates:

$$r_1 = r_{1bl} + r_{cag}[G_s] \frac{[Ca^{2+}]^3}{K_{ca}^3 + [Ca^{2+}]^3} \frac{K_{Gi}}{K_{Gi} + [G_i]}, \tag{8}$$

$$r_2 = r_{2bl} + r_{ca}[Ca^{2+}]. \tag{9}$$

Other equations for the PKA pathway follow the model of cAMP dynamics by Violin et al [48]:

$$\frac{d[cAMP]}{dt} = k_8[AC] - \frac{k_{10}[PDE][cAMP]}{k_m + [cAMP]}, \tag{10}$$

$$\frac{d[PKA]}{dt} = k_{a7}[PKA_i][cAMP] - k_7[PKA], \tag{11}$$

$$\frac{d[PDE]}{dt} = k_{13} + k_{14}[PKA] \frac{[PDE_i]}{k_{m2} + [PDE_i]} - k_{15}[PDE], \tag{12}$$

where $[AC_i]$, $[PKA_i]$ and $[PDE_i]$ are the inactive states. The initial concentrations in $\mu M$ that differ from zero are $[AC_i]$ = 0.1, $[cAMP]$ = 1.9, $[PKA_i]$ = 1, $[AC_i]$ = 5, and $[PDE]$ = 0.1.

## RGS protein

The evolution of RGS protein is given by:

$$\frac{d[RGS]}{dt} = (r_5 + \alpha)(1 - [RGS]) - r_6[RGS] \,, \tag{13}$$

$$\alpha = \begin{cases} 1, & \text{if } z > 0.1 \\ 0, & \text{otherwise} \end{cases} \tag{14}$$

$$\frac{dz}{dt} = r_3(1 - z) - r_4 z \,, \tag{15}$$

$$r_3 = r_{3bl} \frac{[Ca^{2+}]^2}{K_{ca} + [Ca^{2+}]} \,. \tag{16}$$

Initially both z and [RGS] are zero.

## G protein-coupled receptor

As soon as a climbing fiber input arrives during 200 ms generates the following quantity of G alpha protein:

$$[G_s] = \frac{5K_g{}^3}{K_g{}^3 + [RGS]^3} \,. \tag{17}$$

## Supporting information

**S1 Table. Values for all parameters and coefficients of the model.**
(PDF)

## Acknowledgments

The authors thank the Gateway Portal for the computing resources provided [45]. The authors are indebted to Luiz Lana for writing a draft of the introduction, in depth discussions leading to the proposed mechanism and many ideas here discussed. Likewise the authors thank Jaan Aru, Andres Laan and Juhan Aru for fruitful discussions.

## Author Contributions

**Conceptualization:** Daniel Majoral, Ajmal Zemmar, Raul Vicente.

**Funding acquisition:** Raul Vicente.

**Investigation:** Daniel Majoral.

**Methodology:** Daniel Majoral.

**Project administration:** Raul Vicente.

**Resources:** Raul Vicente.

**Software:** Daniel Majoral.

**Supervision:** Raul Vicente.

**Validation:** Daniel Majoral, Ajmal Zemmar.

**Visualization:** Daniel Majoral.

**Writing – original draft:** Daniel Majoral, Ajmal Zemmar, Raul Vicente.

**Writing – review & editing:** Daniel Majoral, Ajmal Zemmar, Raul Vicente.

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
