## [Decision Letter · Decision Letter 0]

25 Oct 2019

Dear Dr Vicente,

Thank you very much for submitting your manuscript, 'A model for time interval learning in the Purkinje cell', to PLOS Computational Biology. As with all papers submitted to the journal, yours was fully evaluated by the PLOS Computational Biology editorial team, and in this case, by independent peer reviewers. The reviewers appreciated the attention to an important topic but identified some aspects of the manuscript that should be improved.

We would therefore like to ask you to modify the manuscript according to the review recommendations before we can consider your manuscript for acceptance. Your revisions should address the specific points made by each reviewer and we encourage you to respond to particular issues Please note while forming your response, if your article is accepted, you may have the opportunity to make the peer review history publicly available. The record will include editor decision letters (with reviews) and your responses to reviewer comments. If eligible, we will contact you to opt in or out.raised.

- Supporting Information uploaded as separate files, titled 'Dataset', 'Figure', 'Table', 'Text', 'Protocol', 'Audio', or 'Video'.

We hope to receive your revised manuscript within the next 30 days. If you anticipate any delay in its return, we ask that you let us know the expected resubmission date by email at ploscompbiol@plos.org.

Sincerely,

Francesco P. Battaglia

Associate Editor

PLOS Computational Biology

Samuel Gershman

Deputy Editor

PLOS Computational Biology

[LINK]

Reviewer's Responses to Questions

**Comments to the Authors:**

Reviewer #1: I thoroughly enjoyed reading this short but clearheaded manuscript.

The paper introduces a novel and insighful sub-cellular model that should account for paradoxical results from eye-blink conditioning experiments.

The introduction is well argued and brings in relevant experiments. The suggestion that calcium oscillations in micro-domains is a component of learning is interesting and timely.

I did miss reference and discussion to recent molecular modelling results about the graded signal of the CS on Purkinje cells [1] which are aligned to the effects of calcium dynamics in the manuscript. Also, I would like to see a discussion of the role of signalling variability (CS duration / CS amplitude / firing frequency / dendritic diameters [as in 2]) on the molecular and circuit mechanisms.

In the discussion (line 211) the authors equate Purkinje cell interstimulus interval learning with learning of eye blink delay. I am unaware (but happy to learn) of studies that have tested minimum interval learning with electrical stimulus. I missed extensive references that would be required to back up with literature the problems 2 and 3.

Also, in the discussion, the (compelling) suggestion that multiple domains learn different intervals, needs to be complemented by a couple of statements how different intervals would not be overwritten (if the PF bundles would be the same).

Finally, concerning the limitations of the model (line 275):

- you are simulating simple and complex spikes but looking at simplified calcium and G protein mechanisms. The model limitations (line 275) should mention that the signals from inferior olive cells are not exactly synchronous nor particularly well timed. What would happen to these results if say, 100 Purkinje cells were connected to 10 Inferior Olive cells with timings that are not precise nor reliable? If these olivary cells are synchronous and coupled, can you still conclude that the "yet to be understood mechanism" resides exclusively in the Purkinje cell?

Typos and grammar:

50: dot missing after 'consolidated'

61: 'in return' -> in turn

85: inverted quotes

133: "When through parallel fibers..." consider direct phrasing

[1] C. G. Zamora Chimal and E. de Schutter, “Ca2+ Requirements for Long-Term Depression Are Frequency Sensitive in Purkinje Cells.,” Front. Mol. Neurosci., vol. 11, p. 438, 2018.

[2] H. Anwar, C. J. Roome, H. Nedelescu, W. Chen, B. Kuhn, and E. de Schutter, “Dendritic diameters affect the spatial variability of intracellular calcium dynamics in computer models.,” Front. Cell. Neurosci., vol. 8, p. 168, 2014.

Reviewer #2: This is an excellent paper! It is clear, succinct, well written. I like the fact that the authors themselves point to limitations and issues that should be explored further.

The paper is acceptable as it is. Nevertheless, I have some very trivial corrections and some suggestions for further thought. The latter should not be taken as requirements for rewriting but suggestions that the authors might find useful in improving the paper.

Minor corrections:

1. Page 1: the sense in which the Purkinje cell CR is timed should be explained. Readers not familiar with the area will not know if it is onset or offset or time to maximum amplitude that is timed.

2. Acronyms should be explained when they first appear (for instance IP3, PKA, PIP3).

3. L 50-51 “based in known” should be “based on known”.

4. L 52 delete “to”.

5. L 163. I do not understand this sentence. Should “with” be “to” instead?

6. Fig. 5 legend: number of previous pairings.

7. When the manuscript is printed out, some figures come out so small I had to use a magnifying glass. I hope this will be better in the final version.

Suggestions for improvement:

1. A crucial principle is the decoupling mechanism that will generate the same clock activity regardless of CS intensity or duration. A very brief CS of a couple of milliseconds has the same effect as a several hundred millisecond long CS (by the way, the correct reference is Jirenhed & Hesslow (2011) J Neuroscience 31:9070 –9074) but it is not clear how this is achieved. The text is to elliptic. I also found the illustration and description of the clock in relation to figure 2 confusing. The figure looks like an oscillating process with no trend, making it mysterious how a coincidence detection could detect how much time has passed. I assume that the clock involves a continuous change, such as increased calcium level, but this section could be expanded and explained more clearly. It might also be explained here why a long CS would not change the activity of the clock (decoupling). I realize that some of this is explained later in the paper but there is a point in introducing the basic principles in abstract terms and I think it can be done better.

2. There are a number of interesting features of conditioning that a good model should be able to explain. In particular I think that the time course of the Purkinje cell CR, the pause response, should be dealt with. Just as with overt CRs, the onset latencies of Purkinje CRs do not vary much with varying CS-US intervals. What is timed is mainly the maximum amplitude and the offset. This is such a basic feature of the CR that it would seem to be a crucial test of the model that it can account for it.

3. Another interesting fact is the change in CR latency that appears when the CS intensity is suddenly increased. Can they explain this intensity effect? (Svensson et al. 1997, Learning & Memory 3:105-115; Svensson et al. 2010, J Neurophysiol 103:1329-1336.)

**Have all data underlying the figures and results presented in the manuscript been provided?**

Reviewer #1: Yes

Reviewer #2: Yes

PLOS authors have the option to publish the peer review history of their article (what does this mean?). If published, this will include your full peer review and any attached files.

Reviewer #1: Yes: Mario Negrello

Reviewer #2: No

---

## [Editor Report · Decision Letter 1]

11 Dec 2019

Dear Dr Vicente,

We are pleased to inform you that your manuscript 'A model for time interval learning in the Purkinje cell' has been provisionally accepted for publication in PLOS Computational Biology.

In the meantime, please log into Editorial Manager at https://www.editorialmanager.com/pcompbiol/, click the "Update My Information" link at the top of the page, and update your user information to ensure an efficient production and billing process.

One of the goals of PLOS is to make science accessible to educators and the public. PLOS staff issue occasional press releases and make early versions of PLOS Computational Biology articles available to science writers and journalists. PLOS staff also collaborate with Communication and Public Information Offices and would be happy to work with the relevant people at your institution or funding agency. If your institution or funding agency is interested in promoting your findings, please ask them to coordinate their releases with PLOS (contact ploscompbiol@plos.org).

Thank you again for supporting Open Access publishing. We look forward to publishing your paper in PLOS Computational Biology.

Sincerely,

Francesco P. Battaglia

Associate Editor

PLOS Computational Biology

Samuel Gershman

Deputy Editor

PLOS Computational Biology

---

## [Editor Report · Acceptance letter]

4 Feb 2020

PCOMPBIOL-D-19-01573R1 

A model for time interval learning in the Purkinje cell

Dear Dr Vicente,

I am pleased to inform you that your manuscript has been formally accepted for publication in PLOS Computational Biology. Your manuscript is now with our production department and you will be notified of the publication date in due course.

With kind regards,

Sarah Hammond
